# Predicting Chemotherapy-Related Adverse Events in Elderly Cancer Patients with Prior Anticancer Therapy

**Hirotaka Suto [1,2,*], Yumiko Inui [2] and Atsuo Okamura [2]**

1   Department of Medical Oncology, The Cancer Institute Hospital of Japanese Foundation for Cancer Research, Tokyo 135-8550, Japan
2   Department of Medical Oncology/Hematology, Kakogawa Central City Hospital, Hyogo 675-8611, Japan; yuinui@med.kobe-u.ac.jp (Y.I.); atsuo@godzilla.kobe-u.ac.jp (A.O.)
*   Correspondence: hirotaka.suto@jfcr.or.jp; Tel.: +81-3-3520-0111

**Abstract:** To test the usefulness of the Cancer and Aging Research Group (CARG) predictive tool, it was used to assess elderly cancer patients with prior anticancer therapy. Among patients with solid malignancies aged ≥ 65 years receiving second-line chemotherapy who were admitted to the Department of Medical Oncology/Hematology at Kakogawa Central City Hospital between April 2016 and September 2019, the risk ≥ grade 3 of developing chemotherapy-related adverse events (CRAEs) (low, intermediate, or high) was calculated using the tool. Correlations between grades 3 and 5 CRAE incidence rates in the first course of each regimen and CARG risk score, age, and Eastern Cooperative Oncology Group performance status (ECOG PS) were assessed. Included patients ($n = 62$) had a mean age of 71 years (range, 65–82 years). Severe CRAE incidence in patients with low, medium, or high CARG risk was 27%, 54%, and 71%, respectively ($p = 0.026$). The incidence of severe non-hematological toxicities was 5%, 35%, and 64%, respectively ($p < 0.01$). There was no association between age or ECOG PS and chemotherapy toxicity. The results suggest the validity of the CARG predictive tool in elderly cancer patients with prior anticancer therapy. Particularly, the tool showed potential for predicting non-hematological toxicity.

**Keywords:** Cancer and Aging Research Group predictive tool; elderly patients; chemotherapy-related adverse events

## 1. Introduction

In the United States, cancer is a disease of the elderly, with approximately 80% of all cancer deaths among individuals aged ≥60 years [1]. Similarly, in Japan, approximately 70% of all cancers are in those aged ≥65 years [2]. As a result, the use of anticancer drug therapy is increasing in the elderly, which increases the risk of chemotherapy-related adverse events (CRAEs) among the elderly population [3,4]. With the newly established use of adjuvant chemotherapy and the development of treatment with various molecular targeted agents and immune checkpoint inhibitors, including biliary tract cancer, CRAEs have also become more varied [5,6]. Nonetheless, few tools that are useful for characterizing chemotherapy-related risks in older patients with solid tumors are in current use.

Regardless of age, Eastern Cooperative Oncology Group performance status (ECOG PS) is used to predict chemotherapy toxicity and the likelihood of survival in patients [7]. However, whether ECOG PS is a valid predictor of toxicity remains unclear. The validity of ECOG PS is especially controversial in the elderly because the measure was validated in young adults, without addressing the health status diversity observed among elderly cancer patients. Therefore, Hurria et al. developed the Cancer and Aging Research Group (CARG) predictive tool, a prediction model for CRAEs of elderly cancer patients [8,9]. The tool facilitates the prediction of grade 3–5 CRAE tumors in elderly patients with solid tumors, which was previously difficult using age and ECOG PS.

However, most patients included in their study were treated with first-line anticancer drugs, and the model was not validated in patients treated with second-line or subsequent anticancer therapies [8,9]. Therefore, it remains unclear whether pre-treatment influences the prediction of adverse events. To test the usefulness of the CARG predictive tool, it was used to assess elderly cancer patients who received prior anticancer therapy.

## 2. Materials and Methods

### 2.1. Study Population

Patients aged ≥65 years with solid tumors who received second-line or subsequent chemotherapy were eligible for inclusion in the study. Patients who visited the Department of Medical Oncology/Hematology of Kakogawa Central City Hospital between April 2016 and September 2019 and received a new patient's anticancer drug regimen were considered. Patients receiving concurrent radiation were excluded.

The study was approved by the institutional review board of Kakogawa Central City Hospital. The study was conducted in accordance with the tenets of the 1963 Declaration of Helsinki and its later versions. Due to the retrospective design of this study, the requirement for patients' informed consent was waived by our hospital's institutional review board.

### 2.2. Study Design

Prior to chemotherapy, patients completed a medical questionnaire that included information regarding the presence of comorbidities, hearing impairment, falls in the past 6 months, walking restriction for 100 m, need for medication assistance, and loss of social activities due to physical and mental health status. In addition, we recorded tumor characteristics (type and stage), pre-treatment laboratory data, line of chemotherapy (second-line or later), use of granulocyte colony-stimulating factor (G-CSF), and first-line chemotherapy drugs and dosage. Chemotherapy dosing for the first cycle of treatment was categorized as either standard or dose-reduced, as per American National Comprehensive Cancer Network Guidelines. Grade 3–5 CRAEs during chemotherapy were defined as per National Cancer Institute Common Terminology Criteria for Adverse Events (NCI-CTCAE), version 5.0, via a medical record review. Laboratory-based toxicities were identified based on laboratory values on the date of scheduled chemotherapy or when patients sought medical care for symptoms that presented between chemotherapy cycles.

### 2.3. Statistical Analysis

A chemotherapy toxicity score was calculated for each patient by using 11 prechemotherapy variables included in the CARG predictive tool for chemotherapy toxicity (Table 1) [8,9]. Chemotherapy toxicity risk was categorized as follows: low (0–5 points), moderate (6–9 points), or high (≥10 points) [8,9]. We evaluated the correlation between the incidence of grade ≥3 CRAEs during the first course of each regimen and CARG risk score or the patient characteristics. Toxicity distributions among different risk groups were compared with the ability of ECOG PS or age to predict toxicity. To make this comparison, ECOG PS scores were divided into three groups (0, 1, and ≥2), and patients were also divided into three age groups (65–69 years, 70–74 years, and ≥75 years).

The Fisher's exact test was used to evaluate between-group differences in the incidence of grade 3–5 toxicity. All statistical analyses were performed using EZR software (Saitama Medical Center, Jichi Medical University, Saitama, Japan) [10]. A $p$-value < 0.05 was considered statistically significant.

**Table 1.** Prediction model and scoring algorithm values used to predict chemotherapy toxicity.

| Variable | Value/Response | Score |
|---|---|---|
| Age | ≥72 years | 2 |
| | <72 years | 0 |
| Cancer type | GI or GU cancer | 2 |
| | Other cancer types | 0 |
| Planned chemotherapy dose | Standard dose | 2 |
| | Dose reduced upfront | 0 |
| Planned number of chemotherapy drugs | Polychemotherapy | 2 |
| | Monochemotherapy | 0 |
| Hemoglobin level | <11 g/dL (male) <10 g/dL (female) | 3 |
| | ≥11 g/dL (male) ≥10 g/dL (female) | 0 |
| Creatinine clearance | <34 mL/min | 3 |
| | ≥34 mL/min | 0 |
| How is your hearing (with a hearing aid if needed)? | Fair, poor, or totally deaf | 2 |
| | Excellent or good | 0 |
| Number of falls in the past 6 months | ≥1 | 3 |
| | None | 0 |
| Can you take your own medicine? | Able with some help/unable | 1 |
| | Able without help | 0 |
| Does your health limit your ability to walk 100 m? | Somewhat limited/limited a lot | 2 |
| | Not limited at all | 0 |
| During the past 4 weeks, how much of the time have your physical health or emotional problems interfered with social activities (like visiting with friends, relatives, etc.)? | Limited some of the time, most of the time, or all of the time | 1 |
| | Limited none of the time or a little of the time | 0 |

Abbreviations: GI, gastrointestinal; GU, genitourinary.

## 3. Results

### 3.1. Patient Characteristics

The study population comprised 62 cancer patients aged ≥65 years (Table 2). The median age of participants was 71 years (range, 65–82 years), and 53% were male. The most common type of cancer was gastrointestinal cancer (58%). Seventy-one percent of all patients received a single agent, and 69% received standard doses of chemotherapy. The number of patients with ECOG PS scores of 0, 1, and 2 was 11, 40, and 11, respectively. When stratified according to the CARG risk score, 22 patients were placed in a low-risk group, 26 in an intermediate-risk group, and 14 in a high-risk group.

**Table 2.** Patient characteristics.

| Patient Characteristic | Number | % |
|---|---|---|
| Age (years) | | |
| 65–69 | 28 | 45 |
| 70–74 | 16 | 26 |
| 75–79 | 14 | 23 |
| ≥80 | 4 | 6 |
| Sex | | |
| Male | 33 | 53 |
| Female | 29 | 47 |
| ECOG PS | | |
| 0 | 11 | 18 |
| 1 | 40 | 64 |
| ≥2 | 11 | 18 |

**Table 2.** *Cont.*

| Patient Characteristic | Number | % |
|---|---|---|
| Cancer type | | |
| Breast | 9 | 15 |
| Lung | 4 | 6 |
| GI | 36 | 58 |
| GYN | 2 | 3 |
| GU | 0 | 0 |
| Other | 11 | 18 |
| Treatment | | |
| Standard dose | | |
| Yes | 43 | 69 |
| No | 19 | 31 |
| Number of chemotherapy drugs | | |
| Monochemotherapy | 44 | 71 |
| Polychemotherapy | 18 | 29 |
| Line of chemotherapy | | |
| Second | 39 | 63 |
| ≥Third | 23 | 37 |
| Growth factor use | | |
| Yes | 7 | 11 |
| No | 55 | 89 |
| Hemoglobin | | |
| <10 g/dL (female) | 16 | 26 |
| ≥10 g/dL (female) | 13 | 21 |
| <11 g/dL (male) | 17 | 27 |
| ≥11 g/dL (male) | 16 | 26 |
| Creatinine clearance | | |
| <34 mL/min | 4 | 6 |
| ≥34 mL/min | 58 | 94 |
| Hearing | | |
| Fair, poor, or totally deaf | 3 | 5 |
| Excellent or good | 59 | 95 |
| No. of falls in the past 6 months | | |
| ≥1 | 2 | 3 |
| None | 60 | 97 |
| Taking medications | | |
| With some help/unable | 5 | 8 |
| Without help | 57 | 92 |
| Limited in walking 100 m | | |
| Somewhat limited/limited a lot | 18 | 29 |
| Not limited | 44 | 71 |
| Decrease in social activity due to health/emotional problems | | |
| Some, most, all of the time | 21 | 34 |
| A little, or none of the time | 41 | 66 |
| CARG | | |
| 0–5 (low) | 22 | 35 |
| 6–9 (intermediate) | 26 | 42 |
| ≥10 (high) | 14 | 23 |

Abbreviations: CARG, Cancer and Aging Research Group; ECOG PS, Eastern Cooperative Oncology Group performance status; GI, gastrointestinal; GU, genitourinary; GYN, gynecologic.

### 3.2. Chemotherapy-Related Adverse Events

The most commonly observed grade 3–5 hematologic toxicities were neutropenia (21%) and leucopenia (15%). Grade 4 hematologic toxicities included neutropenia (8%), leucopenia (2%), and thrombocytopenia (2%). The most commonly observed grade 3–5 nonhematologic toxicities were nausea (18%), fatigue (13%), and oral mucositis (6%). The

grade 4 nonhematologic toxicities included hyponatremia (2%) and hypomagnesemia (2%) (Table 3). No patient died due to CRAEs.

**Table 3.** Chemotherapy-related adverse events.

| Adverse Event | Grade 3–5 CRAE No. % | | Grade 3 CRAE No. % | | Grade 4 CRAE No. % | |
|---|---|---|---|---|---|---|
| Hematologic | | | | | | |
| Leukopenia | 9 | 15 | 8 | 13 | 1 | 2 |
| Neutropenia | 13 | 21 | 8 | 13 | 5 | 8 |
| Anemia | 5 | 8 | 5 | 8 | 0 | 0 |
| Thrombocytopenia | 4 | 6 | 3 | 5 | 1 | 2 |
| Febrile neutropenia | 1 | 2 | 1 | 2 | 0 | 0 |
| Nonhematologic | | | | | | |
| Fatigue | 8 | 13 | 8 | 13 | 0 | 0 |
| Nausea | 11 | 18 | 11 | 18 | 0 | 0 |
| Mucositis oral | 4 | 6 | 4 | 6 | 0 | 0 |
| Diarrhea | 1 | 2 | 1 | 2 | 0 | 0 |
| Hypertension | 1 | 2 | 1 | 2 | 0 | 0 |
| Proteinuria | 1 | 2 | 1 | 2 | 0 | 0 |
| Edema | 1 | 2 | 1 | 2 | 0 | 0 |
| Hyponatremia | 1 | 2 | 0 | 0 | 1 | 2 |
| Hyperkalemia | 2 | 3 | 2 | 3 | 0 | 0 |
| Hypomagnesemia | 1 | 2 | 0 | 0 | 1 | 2 |

Abbreviations: CRAE, chemotherapy-related adverse event; No., number.

*3.3. Comparison of CARG, ECOG PS, and Age for Predicting the Occurrence of Grade 3–5 CRAEs*

As shown in Table 3, CRAE in patients classified as having low, medium, or high CARG risk scores was 27%, 54%, and 71%, respectively ($p = 0.026$) (Figure 1a). When classified based on the patient's ECOG PS score, the incidence of grade 3–5 CRAEs in patients with PS scores of 0, 1, and $\geq$2 was 36%, 48%, and 64%, respectively ($p = 0.50$) (Figure 1b). In addition, the incidence of $\geq$grade 3 CRAEs was 54% for those aged 65–69 years, 44% for those aged 70–74 years, and 44% for those aged $\geq$75 years or older, with no correlation between age and the incidence of severe CRAEs observed ($p = 0.85$) (Figure 1c).

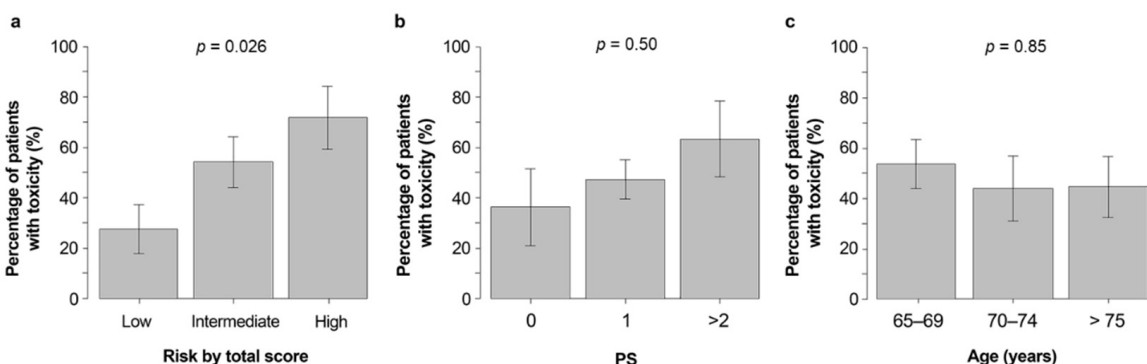

**Figure 1.** Grade 3–5 CRAE incidence in those with low, medium and high (**a**) CARG risk scores, (**b**) PS, and (**c**) age classifications. Abbreviations: CARG, Cancer and Aging Research Group; PS, performance status; Gr3–5 CRAEs, grade 3–5 chemotherapy-related adverse events.

The incidence of severe hematological toxicities was 27%, 27%, and 42% in patients with low, medium, and high CARG risk scores, respectively ($p = 0.60$) (Figure 2a), while the incidence of severe non-hematological toxicities was 5%, 35%, and 64%, respectively ($p < 0.01$) (Figure 2b).

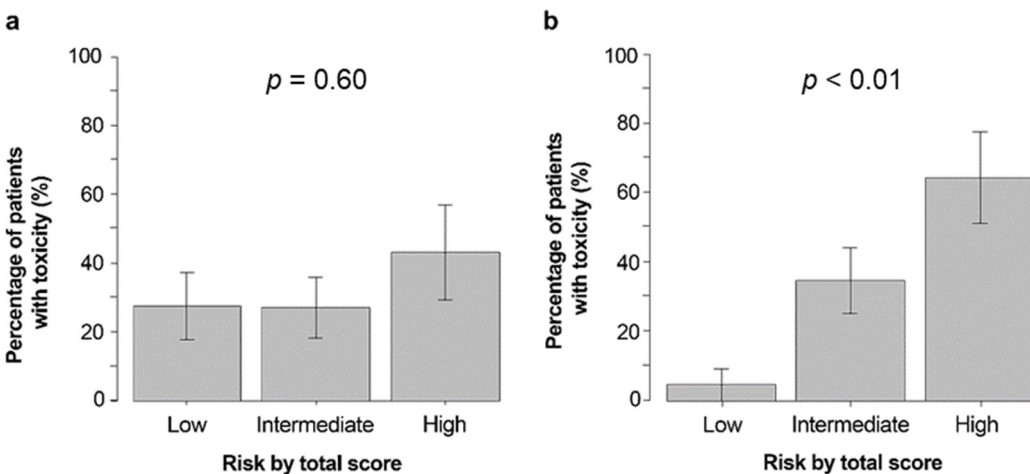

**Figure 2.** Percentage of participants with severe hematologic (**a**) and non-hematologic (**b**) toxicities based on CARG predictive tool risk strata. Abbreviations: CARG, Cancer and Aging Research Group.

## 4. Discussion

In developed countries, cancer incidence and mortality rates are high [11], as is anticancer drug treatment use. In recent years, with the development of anticancer drug therapy, opportunities for the elderly to receive anticancer drug therapy after first-line treatment have increased. However, few methods for predicting CRAEs in elderly patients receiving second-line and subsequent anticancer therapies have been explored [8,9,12–15]. The CARG predictive tool enables the prediction of the occurrence of grade 3–5 CRAEs in elderly cancer patients receiving first-line treatment, which was previously difficult to predict based on age and ECOG PS [8,9]. Therefore, the American Society of Clinical Oncology guidelines recommend the use of this tool [16].

Our study suggests that the CARG tool predicts grade 3–5 CRAEs well in elderly cancer patients who received prior anticancer therapy. In this patient population, it was difficult to predict severe CRAEs based on age or ECOG PS. Another tool for predicting CRAEs is the Chemotherapy Risk Assessment Scale for High-Age Patients (CRASH) score. In the previous study, the CRASH score predicted grade 3–5 CRAEs in cancer patients aged ≥70 years. The CRASH score also allowed the stratification of the risk in hematologic and non-hematologic toxicities [15]. Approximately half of the validation cohort considered, when assessing the ability of CRASH scores to predict CRAEs risk, were patients receiving second-line or subsequent chemotherapy. However, there are no data on CRASH scores and adverse events in previously treated patients only, and it remains unclear whether CRASH scores accurately predict adverse events among this population [15]. Furthermore, the CRASH score does not facilitate the assessment of the toxicity of new treatment.

The current study showed that the CARG risk score correlated more strongly with the incidence of severe non-hematologic toxicity than that of severe hematologic toxicity. Among the study population, a standard anticancer drug dosage rate of approximately 70%, a monotherapy rate of approximately 70%, and a G-CSF use rate of approximately 10% occurred, which might explain the reduced incidence of severe hematologic toxicity observed.

This study had some limitations. First, it was a retrospective study with a relatively small sample size. To produce more generalizable findings, a prospective multicenter study is necessary. Second, the assessment of non-hematologic toxicity might vary among clinicians and nurses more than that of hematologic toxicity. However, since the clinicians involved in this study performed clinical assessments based on NCI-CTCAE, and the number of the clinicians was small, variation was unlikely to be significant. Although a previous study reported that the ECOG PS scores of clinicians, nurses and patients varied, in the present study, clinician evaluations were preferred because they tended to best reflect prognosis [17]. Third, it is currently unknown whether new molecular targeted drugs or

immune checkpoint inhibitors can predict toxicity. Therefore, it is necessary to accumulate data from elderly cancer patients receiving new molecular targeted drugs and immune checkpoint inhibitors to validate the usefulness of the CARG predictive tool.

Importantly, the extent to which anticancer drug dosage should be reduced in groups classified as high risk using the CARG predictive tool remains unknown. The Geriatric Assessment for Patients 70 years and older (GAP70+), and the Elderly Selection on Geriatric Index Assessment (ESOGIA) 08-02 trials, revealed that the dose adjustment of anticancer drugs using geriatric assessment reduced the incidence of CRAEs but did not improve survival [14,18]. Therefore, the results of the current study suggest that the CARG predictive tool is crucial for managing drug side effects in previously treated cancer patients and might help adjust the dose of anticancer drugs according to the CARG risk scores.

Future efforts are required to accumulate detailed toxicity data using electronic patient-reported outcomes and establish more precise toxicity prediction models using artificial intelligence.

## 5. Conclusions

This study suggested the utility of the CARG tool for predicting adverse event risk in elderly cancer patients who received prior anticancer therapy. In particular, this tool may facilitate the prediction of non-hematological toxicity.

**Author Contributions:** Conceptualization, H.S. and A.O.; methodology, H.S. and A.O.; sample collection, H.S, Y.I., and A.O.; validation, H.S, Y.I., and A.O.; writing—original draft preparation, H.S.; writing—review and editing, H.S., Y.I., and A.O.; supervision, A.O. All authors have read and agreed to the published version of the manuscript.

**Funding:** This research received no external funding.

**Institutional Review Board Statement:** The study was conducted in accordance with the Declaration of Helsinki and approved by the Institutional Review Board Kakogawa Central City Hospital Ethics Committee. The approval code for this study is no.2019079.

**Informed Consent Statement:** Due to the retrospective design of this study, the requirement for patients' informed consent was waived by our hospital's institutional review board.

**Data Availability Statement:** Not applicable.

**Acknowledgments:** The authors thank the medical staff of the Department of Medical Oncology/Hematology, Kakogawa Central City Hospital for their support during this study.

**Conflicts of Interest:** The authors declare no conflict of interest.

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
