# Peer review of "Predicting Chemotherapy-Related Adverse Events in Elderly Cancer Patients with Prior Anticancer Therapy"

_curroncol, doi:10.3390/curroncol29040177_

Round 1

Reviewer 1 Report

the article aims to evaluate the usefulness of the CARG score to predict CRAE in patients previously undergone anti-cancer therapy. Despite a small sample, the article is interesting and the methods seem to support the conclusions. It would be interesting to also use the CRASH score for the same purpose, since this score also comprises the chemotherapy agent and allows to stratify the risk in hematologic and non hematologic part.

Author Response

Thank you for reviewing our manuscript.

Reviewer 2 Report

Dear Editor, thank you so much for inviting me to revise this manuscript.

This study addresses a current topic.

The manuscript is quite well written and organized. English could be improved.

Figures and tables are comprehensive and clear.

The introduction explains in a clear and coherent manner the background of this study.

We suggest the following modifications:

  • Introduction section: although the authors correctly included important papers in this setting, we believe a couple of studies regarding the association between age, chemotherapy, and toxicity should be cited within the introduction (PMID: 32487595 ; PMID: 33571059 ), only for a matter of consistency. We think it might be useful to introduce the topic of this interesting study.
  • Methods and Statistical Analysis: nothing to add.
  • Discussion section: Very interesting and timely discussion. Of note, the authors should expand the Discussion section, including a more personal perspective to reflect on. For example, they could answer the following questions – in order to facilitate the understanding of this complex topic to readers: what potential does this study hold? What are the knowledge gaps and how do researchers tackle them? How do you see this area unfolding in the next 5 years? We think it would be extremely interesting for the readers.

However, we think the authors should be acknowledged for their work. In fact, they correctly addressed an important topic, the methods sound good and their discussion is well balanced.

One additional little flaw: the authors could better explain the limitations of their work, in the last part of the Discussion.

We believe this article is suitable for publication in the journal although some revisions are needed. The main strengths of this paper are that it addresses an interesting and very timely question and provides a clear answer, with some limitations.

We suggest a linguistic revision and the addition of some references for a matter of consistency. Moreover, the authors should better clarify some points.

Author Response

Thank you for reviewing our manuscript.

Round 2

Reviewer 1 Report

The report can be accepted in the present version

Reviewer 2 Report

Acceptance.